# *DeepStack*: Deeply Stacking Visual Tokens is Surprisingly Simple and Effective for LMMs

**Lingchen Meng**[1,2*]    **Jianwei Yang**[3*]    **Rui Tian**[1,2]    **Xiyang Dai**[3]
**Zuxuan Wu**[1,2†]    **Jianfeng Gao**[3†]    **Yu-Gang Jiang**[1,2†]

[1]Shanghai Key Lab of Intell. Info. Processing, School of CS, Fudan University
[2]Shanghai Collaborative Innovation Center of Intelligent Visual Computing
[3]Microsoft Corporation

https://deepstack-vl.github.io/

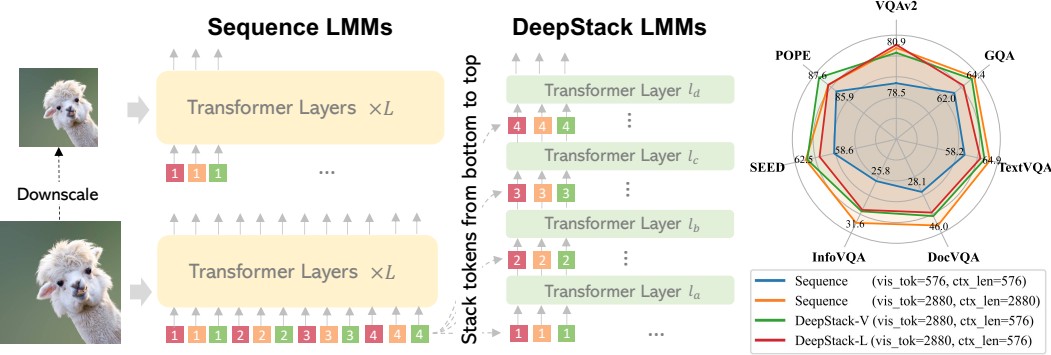

Figure 1: Left: Conventional large multimodal models (LMMs) string all visual tokens into a sequence for high- and low-resolution images. Middle: Our DeepStack LMMs stack the tokens into a grid and infuse them into the first and middle transformer layers from bottom to top (■ ↑ ■ ↑ ■ ↑ ) simply using a residual connection. With *no* architecture modification and context length increasing, our model can handle multiple times more visual tokens as inputs. Right: We apply *DeepStack* separately to Vicuna-7B (DeepStack-L) and CLIP ViT-L (DeepStack-V). Our models can take 4× more visual tokens, and significantly outperforms the sequence LMM with same context length and rival the one using a much longer context, over a wide range of benchmarks.

## Abstract

Most large multimodal models (LMMs) are implemented by feeding visual tokens as a sequence into the first layer of a large language model (LLM). The resulting architecture is simple but significantly increases computation and memory costs, as it has to handle a large number of additional tokens in its input layer. This paper presents a new architecture *DeepStack* for LMMs. Considering $N$ layers in the language and vision transformer of LMMs, we stack the visual tokens into $N$ groups and feed each group to its aligned transformer layer *from bottom to top*, as illustrated in Fig. 1. Surprisingly, this simple method greatly enhances the power of LMMs to model interactions among visual tokens across layers but with minimal additional cost. We apply *DeepStack* to both language and vision transformer in LMMs, and validate the effectiveness of *DeepStack* LMMs with extensive empirical results. Using the same context length, our DeepStack 7B and 13B parameters surpass their counterparts by **2.7** and **2.9** on average across **9** benchmarks, respectively. Using only one-fifth of the context length, DeepStack rivals closely to the counterparts

---

* Equal contributions; † Corresponding authors.

that use the full context length. These gains are particularly pronounced on high-resolution tasks, *e.g.*, **4.2**, **11.0**, and **4.0** improvements on TextVQA, DocVQA, and InfoVQA compared to LLaVA-1.5-7B, respectively. We further apply *DeepStack* to vision transformer layers, which brings us a similar amount of improvements, **3.8** on average compared with LLaVA-1.5-7B.

# 1  Introduction

With the tremendous advancements in large language models (LLMs) [62, 63, 87, 6, 6, 65, 59], we have witnessed a surge of efforts of developing large multimodal models (LMMs) [51, 88]. To connect vision and language models for LMMs, a conventional way is transforming images into a number of visual features using pretrained vision encoders (*e.g.*, CLIP [61]), and flattening them to a sequence of "language tokens" which are then fed into an LLM. With sufficient alignment and instruction tuning, the entire system can demonstrate a broad conversational capability for multimodal inputs [51].

To incorporate visual inputs, it usually requires the LMMs to handle a large number of visual tokens as the prefix tokens in addition to the original language prompts. This inevitably introduces a tremendous memory and compute overhead into the LLMs, which is particularly significant when it comes to high-resolution images and multi-frame videos. Several previous works attempt to mitigate this issue by proposing various token compression strategies. A straightforward way is to reduce the number of tokens with spatial grouping [70, 47]. Instead of pooling vision tokens, a few work instead to concatenate local tokens along the feature dimension to preserve visual information [11, 48]. Moreover, other works seek more sophisticated token resampling, such as Q-Former [43], Perceiver [4] and Abstractor [8], *etc*. In MM1 [57], the researchers performed an extensive analysis of these approaches and found no significant discrepancies among them. Despite the huge effort, all these works inherently sacrifice fine-grained visual information to reach the trade-off between the compute overhead and the information flow into LLMs, which is arguably problematic for high-resolution images and videos. Most recently, a few works [22, 48, 50, 19, 20] proposed multi-crop strategies and string several times more visual tokens to support high-resolution scenarios, while at the cost of substantial overhead.

All current efforts to wire vision with LLMs follow the routine in which visual tokens are always rolled together as a 1d sequence, and fed into the first layer of LLMs as inputs. In this work, we step outside the box and question whether we can find a better strategy to handle the large number of visual tokens regarding both efficacy and efficiency. Instead of examining the LLMs in a traditional left-to-right orientation, we adopt a novel bottom-to-top perspective, revealing that they constitute a hierarchical arrangement of transformer layers. Based on this observation, we propose DeepStack, a simple, yet novel way of feeding visual tokens into LLMs. As shown in Fig. 1, instead of putting the long sequence of visual tokens from left to right, we restructure the visual tokens into a layered stack, where each layer of the stack is connected to one layer in the LLMs by simple residual connection. As a result, with the context length unchanged, we can feed into LLMs several times more visual tokens to handle complex visual inputs. Meanwhile, the combination of per-layer parallel attention and layer-by-layer progression can effectively leverage the LLMs' capacity for modeling the dependencies of visual tokens.

To examine the effectiveness of our method, we apply it to two representative LMMs, LLaVA-1.5 [51] and LLaVA-Next [50]. Extensive empirical results demonstrate the effectiveness of our method. More specifically, with the same setting of LLaVA-1.5, our model can achieve significant performance gain across a wide range of benchmarks. In particular, our model brings **4.2**, **11.0**, and **4.0** performance gains on TextVQA, DocVQA, and InfoVQA compared to LLaVA-1.5-7B, respectively. To summarize, our main contributions are three-fold:

- We propose a simple yet effective *DeepStack* strategy for connecting vision and language in the context of LMMs. This new strategy introduces *no* architecture change while significantly increasing the number of tokens LLMs can take.

- With the *DeepStack* strategy, we present our new model DeepStack, and compare it with LMMs across a wide range of multimodal tasks. Our model demonstrates consistent improvement over the baseline methods, in particular for high-resolution tasks.

- We further conduct comprehensive ablation studies on different aspects of our proposed method, which provide useful guidance and insights behind the design choices.

Finally, although we only demonstrate the effectiveness of our proposed method in the context of LMMs, we note that this simple strategy could be generalized to any models or tasks built on top of transformer layers. We hope this new design could shield new lights and open up new exploratory directions regarding how to wire vision encoders and LLMs in large multimodal models.

## 2 Related Works

**Large Language Models (LLMs).** Recently, natural language processing (NLP) has witnessed significant progress, particularly with the advent of large language models (LLMs) [74, 87, 64, 6]. Building on the foundational architecture of Transformers [75], language models [18, 74, 87, 64, 6, 39] have demonstrated strong scalability through the pretraining-then-finetuning paradigm. Specifically, BERT [18] utilizes the transformer encoder and introduces a masked language modeling task to pre-train the model on vast unlabelled data, showing excellent performance after fine-tuning on downstream tasks. Other follow-ups [39, 36] continue along the lines of BERT, constantly refining and optimizing its performance. The T5 [64] series further unifies different NLP tasks within an encoder-decoder architecture, demonstrating effectiveness across dozens of language understanding tasks. Meanwhile, the GPT [62, 63, 4] series employs simple decoder-only transformers to pretrain the language model using a unified next-token prediction paradigm. This approach shows remarkable scalability in terms of both model size and data scale. To enhance instruction-following abilities, InstructGPT [59] and ChatGPT emphasize the importance of instruction tuning and Reinforcement Learning from Human Feedback (RLHF). These models exhibit excellent capabilities in open-domain conversation tasks, ranging from text generation to question answering. In response to ChatGPT, recent works [74, 15, 38] have made significant efforts in developing an open-source LLMs community. Building on the success of the LLaMA [74] series foundation model, Alpaca [71], Vicuna [15], and GPT-4-LLM [60] showcase the improvements brought by higher-quality instruction datasets. Other works [24, 27, 1, 86] take a different approach, aiming to achieve comparable performance with a much smaller set of parameters. The Phi [24, 27, 1] series revisits the importance of the pre-training corpus and achieves success with models containing around 3 billion parameters. In this paper, we develop our model based on Vicuna [15] and Phi-3 [1], aiming to equip the well-trained LLMs with informative visual tokens and a relatively small training effect.

**Large Multi-modal Models (LMMs).** The success of CLIP [61] and its follow-ups [66, 28, 77] demonstrates the effectiveness of aligning vision and language modalities into a unified semantic space, showcasing promising capabilities in zero-shot classification tasks. More recently, Flamingo [3] and BLIP [44] have utilized visual perceivers [26] to resample visual tokens from image features as inputs for language models through cross-attention. BLIP-2 [42] and Instruct-BLIP [16] further incorporate this mechanism into large language models for tasks such as visual captioning and question-answering. Although visual perceivers can translate image features into a fixed set of visual tokens, they face constraints related to convergence costs and data requirements. In parallel, LLaVA and its follow-ups [13, 76, 47, 50, 49] achieved success in connecting vision and language using a simple projection module. It greatly simplifies the difficulties of alignment tasks and even achieves better performance with less training effort. However, due to the rigorous input resolution of pre-trained models, these directions meet difficulties on downstream tasks requiring finer-grained visual information, *e.g.* tasks relevant to OCR and documents. To alleviate this problem, recent works [48, 22, 21, 73, 89] utilize a mixture of experts (MOE) schemes to leverage different pre-trained vision models, typically assembling the visual tokens along the feature dimension. Other attempts [85, 19, 50] split high-resolution images into multi-crop patches and merge them into a longer sequence, which significantly increases the training and evaluation cost. In this work, we conduct experiments on the projector-based connection framework and revisit the connection scheme that utilizes projected visual tokens for the **input layer** of LLMs. We find that the early layers of LLMs can also well process visual token inputs. Besides that, we propose a *DeepStack* scheme to stack finer-grained visual tokens to the early layers of LLMs, enhancing visual capabilities without introducing extra input tokens.

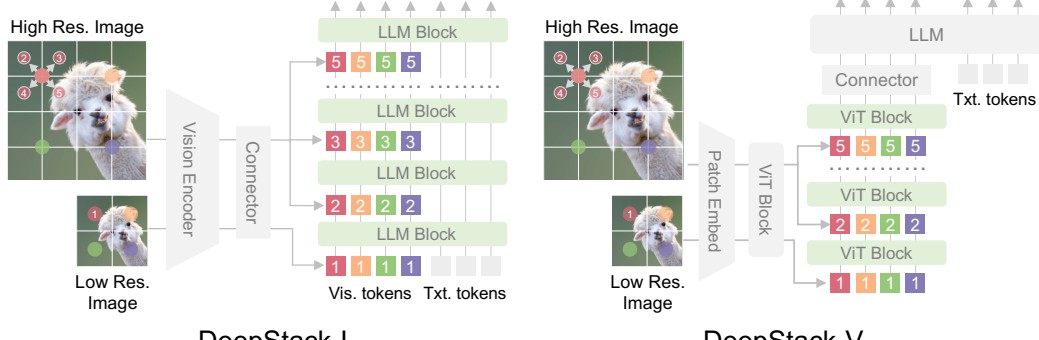

Figure 2: **Architecture of DeepStack.** The main innovation lies in the *DeepStack* strategy that infuses visual tokens into different layers. Left: *DeepStack* for LLMs. Given an input image, we feed the tokens extracted from the low-resolution version to the input layer of LLM. Considering the 2D nature of images, we extra the neighbors from the high-resolution version and reorganize them into *DeepStack*, which are then fed to the consequent layers in LLMs. Right: *DeepStack* for ViTs. We apply similar sampling strategy but feed the visual tokens into the ViT layers of vision encoder.

## 3 DeepStack

*DeepStack* is a versatile strategy that provides finer-grained visual information without increasing the visual context length for LMMs. It achieves this by dividing image feature extraction into two streams: a global-view stream that captures global information, and a high-resolution stream that enhances the global information by stacking dilated high-resolution image features across different layers of the LLMs. This dual-stream approach offers LMMs detailed visual features while maintaining efficiency. By leveraging this simple yet effective method, we build DeepStack, which significantly improves the ability of LMMs to process and comprehend fine-grained visual details. We illustrate DeepStack in Fig. 2 and propose a pseudo-code implementation in Algorithm. 1.

### 3.1 Preliminary: Large Multimodal Model

**Large Language Models (LLMs).** LLMs [2, 11, 70, 74] are typically pre-trained on a huge amount of unlabeled text corpus using a transformer decoder-only architecture. The primary pre-training task is *next-token prediction* driving their learning process. Formally, the learning objective can be formulated as:

$$\mathcal{L} = \sum_{t=1}^{N} \log \mathcal{P}_\theta(x_{t+1} \mid x_{1:t}) \tag{1}$$

where $\mathcal{P}$ represents the large language model and $\theta$ is the trainable parameters of the model, with the training objective to maximize the probability of $x_{t+1}$ as the next token, given the previous tokens $x_{1:t} = x_1, \ldots, x_t$.

**Language Multi-modal Models (LMMs).** LMMs extend pre-trained LLMs to generate responses conditioned on input images. This is achieved by using visual tokens as a prefix:

$$\mathcal{L} = \sum_{t=1}^{N} \log \mathcal{P}_\theta(x_{t+1} \mid x_{1:t}, \mathbf{X}) \tag{2}$$

where $\mathbf{X} \in \mathbb{R}^{l \times c}$ represents the sequence of visual tokens [43, 51, 4], with $l$ being the squence length and $c$ the hidden dimension of the LLM.

**Image Tokenization.** Previous works [45, 43, 51] widely explored how to encode input images into visual tokens. The tokenization schemes usually leverage a vision-language pre-trained image encoder $\mathcal{F}^v$, *e.g.* CLIP [61], to extract image features $\mathbf{f^v}$ from an input image $\mathbf{I}$. Then, the image features are converted into visual tokens using a *connection module* $\mathcal{M}$ as follows:

$$\mathbf{X} = \mathcal{M}(\mathbf{f^v}); \quad \mathbf{f^v} = \mathcal{F}^v(\mathbf{I}) \tag{3}$$

**Algorithm 1:** DeepStack PyTorch pseudocode.

```
# H₀:  Input embeddings for LLM (Original inputs args for traditional LMM);
# vis_pos:  the location of visual tokens;
# X, Xstack:  Original visual tokens, Extra high-resolution visual token list;
# l_start, n:  Index of starting layer, and layer interval for stacking.
1 def forward(H₀, Xstack, l_start, n, vis_pos):
2     H = H₀
3     for (idx, TransformerLayer) in enumerate(self.layers):
          # DeepStack:
4         if idx >= l_start & (idx − l_start)%n == 0:
5             H[vis_pos] += Xstack[(idx − l_start)//n]
          # Original Transformer:
6         H = TransformerLayer(H)
```

The connection module $\mathcal{M}$ can take various forms, mainly divided into projection modules [51, 49] and perceiver resamplers [4, 43]. In the former, $\mathcal{M}$ is implemented as either a single-layer linear projection [51] or a multi-layer MLP [49], directly projecting dense image features into the hidden space of the LLM. In the latter, $\mathcal{M}$ utilizes a cross-attention mechanism with a set of fixed-length learnable queries to extract image features, similar to the approach in [7]. They transform dense image features into sparse image queries, which are then used as input tokens for the language model. However, the resamplers-based methods easily struggle with hallucinations on spatial reasoning tasks [17]. In this paper, we mainly focus on the projection-based connection module for its efficiency and effectiveness.

### 3.2 *DeepStack* for Improved Image Tokenization

Now that we obtain the visual tokens for LMMs using a projection-based connection module, the following challenge is how to provide informative visual tokens while keeping the multi-modal processing effective.

**Scaling Visual Tokens.** Based on the projection-based connection module, many follow-up attempts to increase the visual capability by introducing multiple image crops [50, 73] for scaling up the resolution or involving multiple vision encoders to serve as a mixture of visual experts [89, 73, 21]. For these approaches, the visual tokens from different image crops or vision encoders are concatenated together along the axis of the sequence or the dimension before projection.

***DeepStack* Strategy.** In order to incorporate fine-grained image information while maintaining efficiency, we enhance the input visual tokens $\mathbf{X}$ by stacking high-resolution visual tokens into different LLM decoder layers. In practice, we first upsample the input image according to its aspect ratio and simultaneously tokenize it to obtain high-resolution visual tokens. To prepare the tokens for hierarchy stacking, we split the high-resolution visual tokens into different token sets $\mathbf{X}^{\mathbf{stack}^i}$ with spatial dilation [80, 14]. This sampling approach ensures that the visual tokens $\mathbf{X}^{\mathbf{stack}^i}$ have the same length as the global visual tokens $\mathbf{X}$. Additionally, token $\mathbf{X}^{\mathbf{stack}^i}$ corresponds to the nearest neighbor of $\mathbf{X}$ in spatial.

$$\begin{aligned} \mathbf{X}^{\mathbf{stack}} &= \{\mathbf{X}^{\mathbf{stack}^1}, \mathbf{X}^{\mathbf{stack}^2}, ..., \mathbf{X}^{\mathbf{stack}^s}\} \\ &= \text{Sampling2D}\left(\mathcal{M}(\mathcal{F}^v(\mathbf{I}^{\mathbf{hires}}))\right) \end{aligned} \quad (4)$$

As shown in Fig. 2, given an LLM of $L$ decoder layers, the LLM is first split into different blocks. Specifically, DeepStack split the early layers of LLM $\mathcal{P}$ into a set of *deepstack* blocks $\mathcal{B}^V = \{\mathcal{P}^{V^1}, \mathcal{P}^{V^2}, ..., \mathcal{P}^{V^n}\}$ for stacking visual tokens, and the later layers into a plain block $\mathcal{P}^{\mathbb{L}}$ for original prefix sequential modeling. We denote that each *deepstack* block $\mathcal{P}^{V^i}$ ends at the $N^{V^i}$-th layer of $\mathcal{P}$, while the plain block $\mathcal{P}^{\mathbb{L}}$ ends at the last layer. We use $\mathbf{H}^i$ to represent the hidden states of visual tokens after the $i$-th transformer decoder layer, with $\mathbf{H}^L$ being the visual hidden states after the final decoder layer. Formally, the output of each block can be formulated as follows:

$$\begin{aligned} \mathbf{H}^{V^1} &= \mathcal{P}^{V^1}(\mathbf{X}) + \mathbf{X}^{\mathbf{stack}^1} \\ \mathbf{H}^{V^2} &= \mathcal{P}^{V^2}(\mathbf{H}^{V^1}) + \mathbf{X}^{\mathbf{stack}^2} \\ \mathbf{H}^L &= \mathcal{P}^{\mathbb{L}}(\mathbf{H}^{V^n}) \end{aligned} \quad (5)$$

Specifically, we divide the layers into equally sized *deepstack* blocks, with the block length of 1 by default.

***DeepStack* for Vision Transformers (ViTs).** Our *DeepStack* can be also applied to ViTs for better feature extraction and image tokenization as illustrated in Fig. 2 (DeepStack-V). In contrast to LMM, we use the patch embedding layers $\mathrm{PatchEmbedding}$ and the first several ViT encoder layers for tokenization and the reset ViT encoder layers for *DeepStack*. Formally, we replace the $\mathcal{F}$ and $\mathcal{M}$ in Eq. (4) with the Patch Embedding Layers and the first several encoder layers, and utilize the rest of encoders layers as $\mathcal{P}$ in Eq. (5). Please refer to Sec. 4.3 for more details.

**Comparison with Other Visual Token Enhancement Strategies.** To provide a deeper understanding of the *DeepStack* mechanism, we compare our strategy with previous visual token enhancement strategies by examining the hidden states of visual tokens after the final LLM decoder layer, denoted as $\mathbf{H}^L$. Previous methods can be broadly categorized into two approaches: *Sequence Concatenation* and *Dimension Concatenation*.

As for the former, visual tokens from the entire image and local crops are concatenated sequentially, significantly increasing the overall sequence length the computation cost. The LLM decoder processes these concatenated visual tokens as a longer visual prefix, directly modeling the extended sequence.

$$\mathbf{H}^L = \mathcal{P}\big(\mathrm{SeqCat}[\mathbf{X}, \mathbf{X^{stack}}]\big) \tag{6}$$

As for the latter, visual tokens are concatenated along the feature dimension, keeping the sequence length constant. When using a projection module as the connection module, the enhanced visual tokens can be viewed as the sum of features from two individual projection modules.

$$\begin{aligned}\mathbf{H}^L &= \mathcal{P}\big(\mathcal{M}(\mathrm{DimCat}[\mathbf{f}, \mathbf{f^{hires}}])\big) \\ &\approx \mathcal{P}\big(\mathcal{M}^1(\mathbf{f}) + \mathcal{M}^2(\mathbf{f^{hires}})\big)\end{aligned} \tag{7}$$

In our *DeepStack*, we employ a unique approach where enhancement occurs from bottom to top layer by layer. The processing of $\mathbf{H}^L$ in DeepStack unfolds in two phases. In the early layers of the decoder, the layers function similarly to an encoder, recurrently enhancing the input visual tokens by adding high-resolution visual tokens residually; In the later layers, the decoder performs plain sequence modeling as usual. This dual-phase processing fully leverages the LLM's capabilities by combining both encoding and sequence modeling. By integrating high-resolution visual information at multiple layers, DeepStack effectively enhances visual token representation without increasing visual context length, demonstrating its superiority over previous methods.

Deep layers for LLM sequence modeling

$$\mathbf{H}^L = \mathcal{P}^{\mathbb{L}}\left(\mathcal{P}^{V^n}\left(...\left(\mathcal{P}^{V^1}\big(\mathbf{X} + \mathbf{X^{stack^1}}\big) + \mathbf{X^{stack^2}}\right)...\right) + \mathbf{X^{stack^n}}\right) \tag{8}$$

↑Early layers for visual tokens encoding

## 4 Experiments

### 4.1 Implementation Details

We mainly follow the training recipe of Llava [51], of which the training pipeline consists of two stages, *i.e.* pre-training (PT) stage and supervised-finetuning (SFT) stage. We utilize pre-trained CLIP-large-336 [61] as our default image encoder. To obtain high-resolution feature maps, we split the high-resolution image into patches to comply with the resolution requirement and mosaic the image feature together as whole-image features.

**Pre-training dataset.** We utilize LCS-558k [51] as pre-training data for both experiments based on LLaVA-1.5 and LLaVA-Next, which contain 558k samples from LAION [66], CC [9] and SBU [84], captioned by BLIP [45].

**Fine-tuning datasets.** We utilize LLaVA-mixed-665k [51] as instruction-following data for both experiments based on LLaVA-1.5. However, the SFT dataset used in Llava-Next is not publicly

| Method | LLM | *Eff. Res.* | *Vis. Tok.* | *Cxt. Len.* | PT | SFT | *General VQA* VQA$^{v2}$ | GQA | *Text-oriented VQA* Text VQA$^\ddagger$ | Doc VQA$^\ddagger$ | Info VQA$^\ddagger$ | *LMM benchmarks* SEED (all) | POPE (all) | MM MMU$^\ddagger$ | MM Vet |
|---|---|---|---|---|---|---|---|---|---|---|---|---|---|---|---|
| BLIP-2 [43] | Vicuna-13B | 224 | 32 | 32 | 129M | - | 41.0 | 41.0 | 42.5 | | | 46.4 | 85.3 | - | |
| InstructBLIP [16] | Vicuna-7B | 224 | 32 | 32 | 129M | 1.2M | – | 49.2 | 50.1 | - | - | 53.4 | - | - | - |
| InstructBLIP [16] | Vicuna-13B | 224 | 32 | 32 | 129M | 1.2M | – | 49.5 | 50.7 | - | - | 78.9 | - | - | - |
| Shikra [12] | Vicuna-13B | 224 | - | - | 600K | 5.5M | 77.4* | - | - | - | - | - | - | - | - |
| IDEFICS-9B [37] | LLaMA-7B | 224 | - | - | 353M | 1M | | 50.9 | 38.4 | - | - | - | - | - | - |
| IDEFICS-80B [37] | LLaMA-65B | 224 | - | - | 353M | 1M | 60.0 | 45.2 | - | - | - | - | - | - | - |
| Qwen-VL [5] | Qwen-7B | 448 | 256 | 256 | 1.4B | 50M | 78.8* | 59.3* | 63.8 | - | - | 56.3 | - | - | - |
| Qwen-VL-Chat [5] | Qwen-7B | 448 | 256 | 256 | 1.4B | 50M | 78.2* | 57.5* | 61.5 | - | - | 58.2 | - | - | - |
| VILA [47] | Llama2-7B | 336 | 576 | 576 | 50M | 1M | 79.9* | 62.3* | 64.4 | - | - | 61.1 | 85.5 | - | 34.9 |
| VILA [47] | Llama2-13B | 336 | 576 | 576 | 50M | 1M | 80.8 | 63.3* | 66.6 | - | - | 62.8 | 84.2 | - | 38.8 |
| LLaVA-1.5 [49] | Vicuna-7B | 336 | 576 | 576 | 558K | 665K | 78.5* | 62.0* | 58.2 | 28.1 | 25.8 | 58.6 | 85.9 | 35.3 | 30.5 |
| LLaVA-1.5 [49] | Vicuna-13B | 672 | 576 | 576 | 558K | 665K | 80.0* | 63.3* | 61.3 | 30.3 | 28.4 | 61.6 | 85.9 | 34.8 | 35.4 |
| LLaVA-Next [50] | Vicuna-7B | 672 | 2880 | 2880 | 558K | **765K** | 81.8* | 64.2* | 64.9 | 74.4* | 37.1* | 64.7 | 86.5 | 35.1 | 44.1 |
| LLaVA-Next [50] | Vicuna-7B | 672 | 2880 | 2880 | 558K | **765K** | 82.8* | 65.4* | 66.9 | 77.5* | 44.5* | 65.6 | 86.2 | 35.9 | 49.1 |
| DeepStack-V | Vicuna-7B | 672 | 2880 | 576 | 558K | 665K | 80.4* | 64.1* | 63.5 | 41.0 | 30.0 | 62.3 | 87.6 | 34.9 | 33.0 |
| DeepStack-V | Vicuna-13B | 672 | 2880 | 576 | 558K | 665K | 81.1 | 64.2* | 63.9 | 41.7 | 33.1 | 63.0 | 86.6 | 34.7 | 31.1 |
| DeepStack-L | Vicuna-7B | 672 | 2880 | 576 | 558K | 665K | 79.5* | 63.1* | 62.4 | 39.1 | 29.8 | 60.6 | 86.7 | 35.7 | 29.9 |
| DeepStack-L | Vicuna-13B | 672 | 2880 | 576 | 558K | 665K | 80.9* | 64.2* | 64.6 | 41.5 | 33.0 | 63.5 | 87.7 | 35.2 | 35.9 |
| DeepStack-L-HD† | Vicuna-7B | 1344 | 14400 | 2880 | 558K | 748K | 82.0* | 65.2* | 66.7 | 78.8* | 41.2* | 63.6 | 86.5 | 35.6 | 37.5 |
| DeepStack-L-HD† | Vicuna-13B | 1344 | 14400 | 2880 | 558K | 748K | 83.0* | 66.2* | 68.7 | 81.0* | 45.2* | 65.1 | 86.7 | 33.4 | 39.3 |

Table 1: **Comparison with other LMMs on 9 benchmarks**. *Eff. Res.* indicates the effective image resolution taken by each method. *Vis. Tok.* indicates the number of visual tokens used for LLMs (**not only** for the input layers); *Cxt. Len.* indicates the input visual context length of LLMs. Previous methods feed the visual tokens as the input embeddings, thus the *Vis. Tok.* = *Cxt. Len.* all the time. For comparison with LLaVA-Next, since 765K instruction tuning data is not available, our model is fine-tuned on our 748K data. † indicates that our model is fine-tuned from LLaVA-Next. ∗ The training images of the datasets are observed during training. $^\ddagger$ denotes we report the performance on validation sets. We unfreeze the vision encoder in DeepStack-V and DeepStack-L-HD while freezing it in DeepStack-L for a fair comparison with previous methods. We fine-tuning the vision encoder can bring further improvement on DeepStack-L (please refer to Sec. 4.3 and Supp.)

available, we thus combine an SFT dataset of 748K samples following the guidance [50]. In contrast, we do not involve the user images uploaded to their website.

**Training configuration.** We train our model with only the projection model tuned in the PT stage. In SFT stage, we unfreeze LLM. For Experiments on DeepStack-V and DeepStack-HD, we tune the image encoder with a learning rate of 1e-6 following [50]. Otherwise, we freeze our vision encoder for a fair comparison. We use 16× V100 for experiments with Phi-3 [1] and 8× H100 for experiments with Vicuna [15]. Please refer to our supplementary material for more detailed training hyper-parameters.

## 4.2 Quantitive Results

We evaluate DeepStack on a range of benchmarks, encompassing both academic task-oriented evaluations and recent large multi-modal language model (LMM) benchmarks. Specifically, we focus on text-oriented datasets, including ChartVQA [54], DocVQA [56], InfoVQA [55], MultiDocVQA [72], TextVQA [69], to demonstrate effectiveness in high-resolution scenarios. Additionally, we perform zero-shot evaluations of DeepStack on commonly used video understanding benchmarks to assess its performance on finer-grained tasks.

**General VQA and LMM benchmarks.** We assess DeepStack on two classic general VQA benchmarks, VQAv2 [23] and GQA [25], as well as five recent LMM benchmarks: SEED [40], POPE [46], MMMU [83], and MM-Vet [81]. As presented in Tab. 1, DeepStack outperforms its direct baseline model, LLaVA, on both VQAv2 and GQA, showcasing state-of-the-art performance in traditional VQA tasks. Furthermore, DeepStack consistently surpasses other methods on the recent LMM benchmarks. DeepStack achieves comparable performance on MM-Vet on the experiments based on LLaVA-1.5. However, due to we lack of fancy instruction-following data used in LLaVA-mix-765K, our experiments with LLaVA-Next lag behind the LLaVA-Next. Notably, the significant performance boost on the POPE benchmark suggests that our *DeepStack* strategy effectively alleviates visual hallucination by providing rich and detailed visual information for visual understanding.

**Text-Oriented benchmarks.** To further validate the effectiveness of DeepStack, we evaluate it on more text-oriented benchmarks, including ChartQA [54], DocVQA [56], InfoVQA [55], Multi-DocVQA [72], and TextVQA [69]. These benchmarks contain high-resolution images and typically require the model to answer questions based on fine-grained visual inputs. As shown in Tab. 2,

| Method | LLM | Vis. Tok. | Cxt. Len. | PT | SFT | Chart QA‡ | Doc VQA‡ | Info VQA‡ | MultiDoc VQA‡ | Text VQA‡ |
|---|---|---|---|---|---|---|---|---|---|---|
| LLaVA-1.5 [49] | Vicuna-7B | 576 | 576 | 558K | 665K | 18.2 | 28.1 | 25.8 | 16.7 / 7.2 | 58.2* |
| LLaVA-1.5 [49] | Vicuna-13B | 576 | 576 | 558K | 665K | 18.2 | 30.3 | 29.4 | 18.3 / 8.0 | 61.2* |
| LLaVA-Next [50] | Vicuna-7B | 2880 | 2880 | 558K | **765K** | 54.8 | 74.4 | 37.1 | 44.4 / 31.3 | 64.9 |
| LLaVA-Next [50] | Vicuna-13B | 2880 | 2880 | 558K | **765K** | 62.2 | 77.5 | 44.5 | 46.3 / 32.6 | 66.9 |
| DeepStack-V | Vicuna-7B | 2880 | 576 | 558K | 665K | 20.6 | 41.0 | 30.0 | 23.0 / 11.0 | 63.5* |
| DeepStack-V | Vicuna-13B | 2880 | 576 | 558K | 665K | 20.2 | 41.7 | 33.1 | 23.5 / 11.2 | 63.9* |
| DeepStack-L | Vicuna-7B | 2880 | 576 | 558K | 665K | 19.2 | 39.1 | 29.8 | 22.2 / 10.5 | 62.4* |
| DeepStack-L | Vicuna-13B | 2880 | 576 | 558K | 665K | 19.7 | 41.5 | 33.0 | 22.6 / 10.8 | 64.6* |
| DeepStack-HD† | Vicuna-7B | 14400 | 2880 | 558K | **748K** | 56.3 | 78.8 | 41.2 | 48.2 / 37.7 | 66.7 |
| DeepStack-HD† | Vicuna-13B | 14400 | 2880 | 558K | **748K** | 64.0 | 81.0 | 45.2 | 49.4 / 39.1 | 68.7 |

Table 2: **Results on Text-Oriented benchmarks**, where high resolution is essential. * denotes we use OCR tokens for TextVQA following LLaVA-1.5 [49]. ‡ denotes we report the performance on validation sets.

equipping our model with DeepStack results in consistent gains across all benchmarks. This strongly demonstrates that DeepStack enhances visual token even without increasing sequence length.

**Zero-shot performance on Video QA benchmarks.** We also conduct zero-shot evaluations on video QA benchmarks, including EgoSchema [52] and Next-QA [78] for multiple-choice VQA, and MSVD-QA [10, 79] and ActivityNet-QA [82] for open-ended VQA. Inspired by [33], we sample frames from each video uniformly and mosaic the frames into images to adapt video QA tasks to the image domain. Thanks to the higher effective resolution brought by refined visual tokens, DeepStack effectively handles zero-shot video QA tasks even without being fine-tuned on any video data.

| Method | Multi-choice VQA | | | | | Open-ended VQA | | | |
|---|---|---|---|---|---|---|---|---|---|
| | EgoSchema | Next-QA | | | | MSVD | | ActivityNet | |
| | | Cas. | Des. | Tem. | Acc | Acc. | Score | Acc. | Score |
| LLaVA-1.5-7B | 35.4 | 59.5 | 68.9 | 55.5 | 59.6 | 75.5 | 4.0 | **48.6** | **3.2** |
| DeepStack-L-7B | **38.4** | **61.9** | **69.4** | 55.5 | **61.0** | **76.0** | 4.0 | 49.3 | 3.1 |

Table 3: **Zero-shot evaluation on Video QA benchmarks**. We collate 6 frames uniformly sampled from each video into $2 \times 3$ grid and resize the resulting image to sauare. Our model clearly outperforms the baseline because more visual information is included with the same context length. We mark the best performance **bold**.

## 4.3  Model Inspection

We further conduct sufficient experiments to give in-depth inspiration on the mechanism of DeepStack. In this section, we experiment with phi-3 [1] as the language backbone for the training efficiency. We report the performance on 7 benchmarks, including 1 general VQA (GQA), 2 multi-modal benchmarks (POPE and SEED), and 4 text-oriented VQA (TextVQA, DocVQA, ChartQA and InforVQA). We can evaluate the model performance by comparing the average scores over the 7 benchmarks.

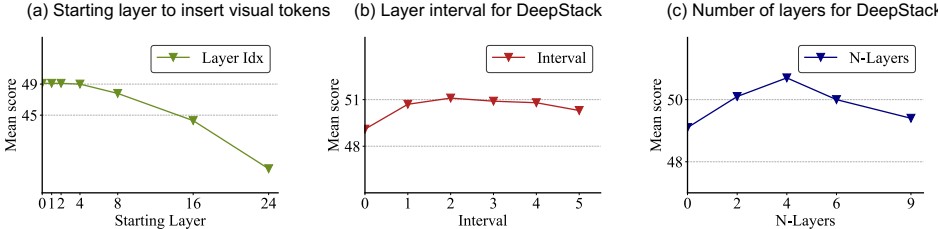

Figure 3: **Analysis on using LLM layers to process visual tokens.** (a) We insert the visual tokens into different starting layers and initialize the correspondence input embeddings as zero; (b) We fix the first layer to insert global visual tokens and ablation on the interval $s$ for stacking high-resolution tokens; (c) We ablation number of layers for token stacking.

**LLMs can well process visual tokens in the early decoder layers.** To understand why earlier layers of LLMs are suitable for processing visual tokens, we conducted an experiment on the insertion layer for visual tokens. Traditionally, visual tokens are inserted at the input layer, *e.g.* 0-th layer. We progressively insert them deeper, initializing the corresponding input embeddings to

zero. As shown in Fig. 3 (a), inserting visual tokens before the 8th of 32 decoder layers in Phi-3 results in acceptable performance variations. However, inserting them beyond the midpoint leads to a significant performance drop. This confirms that earlier layers efficiently handle initial visual information integration. We also explore the impact of inserting visual tokens at non-consecutive layers. In Fig. 3 (b), we fixed global visual tokens at the input layer and varied the interval between two decoder layers for stacking high-resolution tokens. All stacking settings consistently improved performance. Finally, we explored the number of layers used for stacking high-resolution tokens. As shown in Fig. 3 (c), increasing the layers for stacking consistently enhances overall performance, with the best results achieved using four layers.

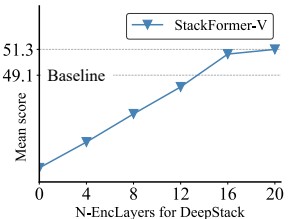

***DeepStack* can also boost Vision Transformers (ViT).** To further explore the potential of DeepStack for vision transformers, we utilize the DeepStack on ViT. Specifically, we use the patch embedding layers and the first $N$ ViT encoder layers to extract visual tokens, including the original tokens and $4\times$ extra high-resolution tokens, and then stack the high-resolution tokens into the next 4 encoder layers, respectively. We need to unfreeze the vision encoder to adapt the pre-trained encoder to our DeepStack. As shown in Tab. 4 and Sec. 4.3, when using the first 16 ViT encoder layers (total 24 layers for our ViT-Large) to extract visual tokens before *DeepStack*, DeepStack-V surpass the baseline model. And the performance keeps increasing when using more encoder layers before *DeepStack*.

| Tok. Enhance | N Layers before *DeepStack* | Ft Enc. | GQA | POPE | SEED | TextVQA | DocVQA | ChartQA | InfoVQA | AVG |
|---|---|---|---|---|---|---|---|---|---|---|
| None | None | | 62.5 | 85.5 | 63.5 | 56.7 | 31.7 | 15.8 | 28.3 | 49.1 |
| None | None | ✓ | 62.4 | 85.8 | 64.0 | 56.1 | 27.5 | 15.3 | 28.3 | 48.5 |
| *DeepStack*-V | PatchEmbed+0 Enc. Layers | ✓ | 56.9 | 80.8 | 54.9 | 44.4 | 13.7 | 12.3 | 25.3 | 41.2 |
| *DeepStack*-V | PatchEmbed+4 Enc. Layers | ✓ | 58.7 | 83.1 | 57.4 | 48.2 | 17.0 | 13.2 | 26.1 | 43.4 |
| *DeepStack*-V | PatchEmbed+8 Enc. Layers | ✓ | 60.4 | 84.2 | 59.7 | 51.8 | 23.1 | 14.7 | 26.6 | 45.8 |
| *DeepStack*-V | PatchEmbed+12 Enc. Layers | ✓ | 61.8 | 85.5 | 62.1 | 55.5 | 29.3 | 16.0 | 26.2 | 48.1 |
| *DeepStack*-V | PatchEmbed+16 Enc. Layers | ✓ | 62.9 | 86.3 | 63.9 | 59.1 | 36.9 | 18.2 | 29.3 | 50.9 |
| *DeepStack*-V | PatchEmbed+20 Enc. Layers | ✓ | 62.8 | 86.1 | 64.0 | 60.1 | 38.4 | 17.1 | 30.6 | 51.3 |

Table 4: **Ablations on the number of ViT encoder layers for DeepStack-V.**

**Better spatial consistency leads to better performance.** Different sampling strategies may lead to different results. In Tab. 5, we compare our default strategy with two other variants for organizing the visual tokens. As shown in Fig. 4, *2d Grid* use each of the local crop as a layer and *1d Sequence* simply flatten the visual tokens to one-dimensional and then re-shape them into a layer stack. Accordingly, keeping the spatial coherence, *i.e. 2d Spatial*, as in our default setting could achieve the best result.

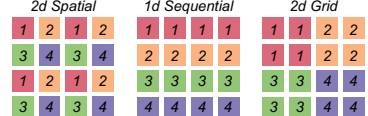

Figure 4: **Visualization of three sampling methods for *DeepStack*.**

| Consistent | Sampling | GQA | POPE | SEED | TextVQA | DocVQA | ChartQA | InfoVQA | AVG |
|---|---|---|---|---|---|---|---|---|---|
| None | None | 62.5 | 85.5 | 63.5 | 56.7 | 31.7 | 15.8 | 28.3 | 49.1 |
| ✗ | *2d Spatial* | 62.2 | 85.1 | 62.3 | 58.1 | 35.1 | 16.4 | 30.1 | 49.9 |
| ✓ | *2d Spatial* | 63.0 | 86.4 | 62.9 | 58.8 | 38.7 | 17.2 | 30.8 | 51.1 |
| ✓ | 2d Grid | 60.6 | 86.2 | 61.2 | 57.1 | 33.2 | 16.4 | 28.6 | 49.0 |
| ✓ | 1d Sequential | 61.6 | 86.2 | 61.9 | 57.1 | 33.1 | 15.2 | 30.0 | 49.3 |

Table 5: **Ablations on image consistency and sampling method**. We apply the Resize transformation to both the original image and the high-resolution image for consistency. For inconsistency, we use Resize on the original image and Pad-Resize on the high-resolution image. *2d Spatial* refers to sampling based on spatial locations, such as using a 4-neighbor method. *2d Grid* means the visual tokens are divided into 2d grids, with each grid stacked per layer. *1d Sequential* indicates that the high-resolution visual tokens are first flattened into a sequence and then uniformly sampled for each layer. Please refer to Fig. 4 for better understanding.

***DeepStack* boosts LMMs from high-resolution tokens, not residual connections**. We experiment to assess the impact of high-resolution images and residual connections in DeepStack by stacking original visual tokens into different layers. As shown in Tab. 6, stacking repeated original tokens (dummy tokens) does not improve performance. This indicates that the performance boost in DeepStack comes from the high-resolution tokens, not from the residual connections.

| Tok. Enhance | Stack Tok. | GQA | POPE | SEED | TextVQA | DocVQA | ChartQA | InfoVQA | AVG |
|---|---|---|---|---|---|---|---|---|---|
| None | None | 62.5 | 85.5 | 63.5 | 56.7 | 31.7 | 15.8 | 28.3 | 49.1 |
| *DeepStack* | Dummy | 62.2 | 85.3 | 63.8 | 56.9 | 31.2 | 15.4 | 28.8 | 49.1 |
| *DeepStack* | Hi-Res | 63.0 | 86.4 | 62.9 | 58.8 | 38.7 | 17.2 | 30.8 | 51.1 |

Table 6: **Ablations on high-resolution visual tokens for stacking.** Dummy refers to repeating the original visual tokens for token stacking; Hi-Res is our default setting that uses high-resolution visual tokens for stacking.

*DeepStack* **achieves a better trade-off between performance and effectiveness.** We compare Deep-Stack with other token enhancement strategies, including dimension-wise concatenation, sequence-wise with high-resolution visual tokens, and string both global visual and high-resolution tokens. As shown in Tab. 7, although string-based methods can bring significant improvement on some benchmarks, they increase the number of tokens at the same time, which will increase the training and inference cost. Meanwhile, DeepStack achieves the best trade-off between performance and effectiveness without introducing extra visual tokens.

| Tok. Enhance | N Tok. | Eff. Tok. | GQA | POPE | SEED | TextVQA | DocVQA | ChartQA | InfoVQA | AVG |
|---|---|---|---|---|---|---|---|---|---|---|
| None | 576 | 576 | 62.5 | 85.5 | 63.5 | 56.7 | 31.7 | 15.8 | 28.3 | 49.1 |
| Dimension Concat | 576 | 2880 | 59.5 | 86.3 | 62.9 | 56.4 | 35.9 | 16.4 | 28.5 | 49.4 |
| Hi-Res String | 2304 | 2304 | 61.8 | 86.2 | 62.1 | 55.0 | 43.5 | 16.2 | 30.4 | 50.7 |
| Global+ Hi-Res String | 2880 | 2880 | 62.3 | 86.4 | 62.6 | 54.7 | 43.3 | 16.7 | 31.2 | 51.0 |
| *DeepStack* | 576 | 2880 | 63.0 | 86.4 | 62.9 | 58.8 | 38.7 | 17.2 | 30.8 | 51.1 |

Table 7: **Ablations on different token enhancement strategies.** Dimension Concat refers to concatenate $\mathbf{X}$ and $\mathbf{X}^{\text{stack}}$ via the channel of features hidden space; Hi-Res String and Global+Hi-Res String refers to string $\mathbf{X}^{\text{stack}}$ and $[\mathbf{X}, \mathbf{X}^{\text{stack}}]$ via sequence, respectively.

*DeepStack* **unleashes the power after fine-tuning the image encoder.** We further experiment with how DeepStack compared coporated with fine-tuning backbones. As shown in Tab. 4, DeepStack achieves the best performance when fine-tuning the backbone. It is worth noticing that when fine-tuning the backbone without DeepStack, the improvement is limited. After combining backbone fietuning with DeepStack, the performance significantly increases among different benchmarks. It is because of the deep interaction between visual tokens and the LLM decoder.

| Tok. Enhance | Ft Enc. | GQA | POPE | SEED | TextVQA | DocVQA | ChartQA | InfoVQA | AVG |
|---|---|---|---|---|---|---|---|---|---|
| None | | 62.5 | 85.5 | 63.5 | 56.7 | 31.7 | 15.8 | 28.3 | 49.1 |
| None | ✓ | 62.4 | 85.8 | 64.0 | 56.1 | 27.5 | 15.3 | 28.3 | 48.5 |
| *DeepStack* | | 63.0 | 86.4 | 62.9 | 58.8 | 38.7 | 17.2 | 30.8 | 51.1 |
| *DeepStack* | ✓ | 63.1 | 86.8 | 63.9 | 61.1 | 41.2 | 18.9 | 31.5 | 52.4 |

Table 8: **Ablations on fine-tuning vision encoder.** DeepStack achieves best performance after fine-tuning vision encoder.

# 5 Conclusion

In this work, we had presented DeepStack, a simple yet effective way to connect vision and language in the context of LMMs. Unlike previous works that always string (compressed) visual tokens into a sequence, we alternatively introduced a new perspective on transformer decoder layers in LLMs, and proposed a *DeepStack* strategy to feed different visual tokens into different layers of LLMs. This strategy significantly mitigates the efficiency overhead introduced by visual tokens and makes it possible to convey more visual information to LLMs. As a result, our DeepStack demonstrated consistent improvements over two baseline models across a wide range of benchmarks. The benefits are particularly significant on tasks that inherently require more tokens, such as high-resolution image understanding. We hope this new *DeepStack* strategy could open up new ideas on how to connect vision and language for faster and better multimodal models in the regime of LMMs.

**Limitation and Future Works**. DeepStack simply inserts the visual tokens into middle LLMs layers via a residual connection in a heuristic manner. Though it already exhibits promising results, we may find a more powerful way to infuse the visual information, *e.g.*, through gated function or layer-wise positional embeddings. Meanwhile, how to systematically decide the starting layer and number of layers also deserves more study. We leave these as promising directions.

**Acknowledgement** This project was supported by NSFC under Grant No. 62102092.

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

# A  Training Details

## A.1  Custom Supervised Finetuning Dataset

We follow LLaVA-Next [50] to combine a custom data mixture containing 748K SFT data shown in Tab. 9. Following [51, 50], our 748K training data mixture contains (1) LLM instruction following data, *e.g.* ShareGPT [68]; (2) GPT4/GPT4V generated data, *e.g.* LLaVA-instruct [51], ShareGPT4V [13], LAION-GPT4V [35]; (3) academic-task-oriented data, *e.g.* VQAv2 [23], GQA [25], *etc*.

| Dataset | Size | Task Prompt |
|---|---|---|
| ShareGPT [68] | 40K | |
| LLaVA-instruct [51] | 158K | |
| ShareGPT4V [13] | 39K | |
| LAION-GPT4V [35] | 11K | |
| VQAv2 [23] | 83K | |
| GQA [25] | 72K | |
| OKVQA [53] | 9K | |
| OCRVQA [58] | 80K | |
| ChartQA [54] | 7K | "Answer the question using a single word or phrase." |
| DVQA [29] | 16K | |
| DocVQA [56] | 10K | |
| AI2D [31] | 2K | |
| SynthDog-EN [32] | 20K | |
| A-OKVQA [67] | 66K | "Answer with the option's letter from the given choices directly." |
| RefCOCO [30] | 48K | "Provide a short description for this region." |
| VG [34] | 86K | "Provide the bounding box coordinate of the region this sentence describes" |

Table 9: Data combination of our 748K SFT data.

## A.2  Detailed Training Configuration

We list the detailed training hyper-parameters as follows. For evaluation, we utilize LLMs-Eval [41] for evaluation on several benchmarks.

| Hypter-param | PT | DeepStack SFT | DeepStack-V SFT | DeepStack-HD SFT |
|---|---|---|---|---|
| global batch size | 256 | 128 | 128 | 128 |
| lr | 1e-3 | 2e-5 | 2e-5 | 2e-5 |
| backbone lr | freeze | freeze | 2e-6 | 2e-6 |
| lr schedule | cosine decay | cosine decay | cosine decay | cosine decay |
| lr warmup ratio | 0.03 | 0.03 | 0.03 | 0.03 |
| epoch | 1 | 1 | 1 | 1 |
| optimizer | AdamW | AdamW | AdamW | AdamW |

Table 10: Training hyper-parameters.

# B  More Experiments

## B.1  Improved DeepStack-L with Fintuning Vision Encoder

As shown in Tab. 11, after finetuning the vision encoder, our DeepStack-L achieves further improvement. This further demonstrates the effectiveness and the potential of our *DeepStack* strategy.

| Method | LLM | Eff. Res. | Vis. Tok. | Cxt. Len. | PT | SFT | General VQA | | Text-oriented VQA | | | LMM benchmarks | | | |
|---|---|---|---|---|---|---|---|---|---|---|---|---|---|---|---|
| | | | | | | | VQA$^{v2}$ | GQA | Text VQA$^\ddagger$ | Doc VQA$^\ddagger$ | Info VQA$^\ddagger$ | SEED (all) | POPE (all) | MM MU$^\ddagger$ | MM Vet |
| LLaVA-1.5 [49] | Vicuna-7B | 336 | 576 | 576 | 558K | 665K | 78.5* | 62.0* | 58.2 | 28.1 | 25.8 | 58.6 | 85.9 | 35.3 | 30.5 |
| DeepStack-L | Vicuna-7B | 672 | 2880 | 576 | 558K | 665K | 79.5* | 63.1* | 62.4 | 39.1 | 29.8 | 60.6 | 86.7 | 35.7 | 29.9 |
| DeepStack-L⋆ | Vicuna-7B | 672 | 2880 | 576 | 558K | 665K | 81.1* | 63.9* | 64.5 | 39.3 | 30.1 | 63.3 | 86.7 | 37.1 | 29.8 |
| LLaVA-1.5 [49] | Vicuna-13B | 336 | 576 | 576 | 558K | 665K | 80.0* | 63.3* | 61.3 | 30.3 | 28.4 | 61.6 | 85.9 | 34.8 | 35.4 |
| DeepStack-L | Vicuna-13B | 672 | 2880 | 576 | 558K | 665K | 80.9* | 64.2* | 64.6 | 41.5 | 33.0 | 63.5 | 87.7 | 35.2 | 35.9 |
| DeepStack-L⋆ | Vicuna-13B | 672 | 2880 | 576 | 558K | 665K | 82.1* | 65.1* | 65.2 | 43.1 | 34.0 | 64.4 | 86.6 | 34.7 | 36.2 |

Table 11: Improved DeepStack-L with fintuning vision encoder. ⋆ indicates that the vision encoder is fine-tuned.

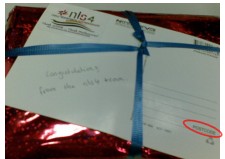 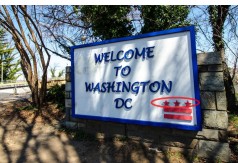 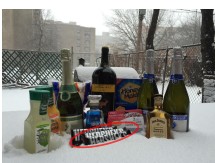 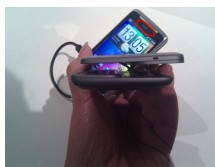

| Q: What does it say in the bottom right corner? | Q: How many stars can you see on the white board? | Q: What candy bar is down there on the bottom? | Q: What brand is the mobile phone? |
|---|---|---|---|
| **LLaVA-1.5**: 'Congratulations' | **LLaVA-1.5**: 1 | **LLaVA-1.5**: Honey maid | **LLaVA-1.5**: Samsung |
| **DeepStack**: 'Postcode' | **DeepStack**: 3 | **DeepStack**: Hershey's | **DeepStack**: HTC |

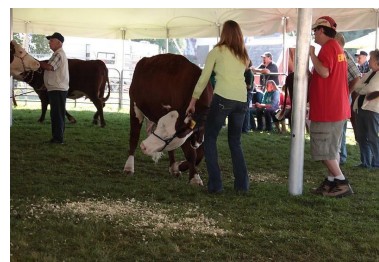

**Q:** Please describe in as much detail as possible what each person in the picture is doing.

**LLaVA-1.5:**
In the image, there is a group of people gathered around a cow in a grassy field. A woman is standing next to the cow, possibly petting it or tending to it. Another woman is standing nearby, possibly observing the scene or waiting for her turn to interact with the cow.There are several other people in the background, some of whom are standing closer to the cow, while others are further away. A few individuals are wearing ties, which suggests a more formal or professional setting. In the scene, there are also a couple of chairs and a dining table, indicating that this gathering might be a social event or a meeting. A handbag can be seen placed on the ground, possibly belonging to one of the attendees.

**DeepStack:**
In the image, there is a group of people gathered around a brown and white cow. A woman is standing next to the cow, holding its halter, while a man is standing nearby, possibly observing the scene. Other people are also present in the area, some standing closer to the cow and others further away. There are a few chairs scattered around the scene, with one near the center and two others closer to the right side. A truck can be seen in the background, possibly indicating that the event is taking place at a fair or similar outdoor venue.

**Q**: Please describe the image as detailed as possible

**LLaVA-1.5:**
The image displays a graph with a central circle that represents a VOQA (Voice over QAM) system. The circle is divided into four quadrants, each representing a different aspect of the system. The quadrants are labeled with the acronyms VOQA, POP, GAOA, and TEX. The graph also includes several smaller circles scattered around the main circle, which are likely related to the VOQA system as well. These smaller circles are positioned at various distances from the main circle, indicating the different levels of the system.

**DeepStack:**
The image features a circular diagram with various labels and numbers. The diagram is divided into different sections, each with its own set of labels and numbers. The central section of the diagram is labeled "VQA2," while the surrounding sections are labeled "Pope," "GQA," "Seed," "InfoVQA," "DocVQA," and "Sequence." There are several numbers in the diagram, with some located in the central section and others scattered throughout the different sections. The numbers are likely used to represent data or information related to the various labels and sections of the diagram.

Figure 5: **Visualization.** Both LLaVA-1.5 and DeepStack use 576 visual context length for a fair comparison. Top: We mark the area corresponding to each question with a **red circle**. DeepStack can well answer the questions which need high-resolution and fine-grained understanding. Bottom: DeepStack demonstrates a more accurate visual understanding in detailed visual captioning.

