# OpenReview forum: "DeepStack: Deeply Stacking Visual Tokens is Surprisingly Simple and Effective for LMMs"
_NeurIPS.cc/2024/Conference — NeurIPS 2024 poster_

### Official Review · Reviewer_REjR · 2024-07-07

**Soundness:** 2
**Presentation:** 2
**Contribution:** 3
**Rating:** 6
**Confidence:** 3

**Summary:**

The authors present a new architecture StackFormer - an effective and simple way to infuse fine-grained visual tokens from CLIP vision transformer to the early layers of  LLaVA-1.5 and LLaVA-Next language models, without increasing the sequence length of visual tokens for LLMs. It doesn't require architecture change while significantly increases the number of tokens LLMs can take, so it improves accuracy especially for high-resolution images and videos.

**Strengths:**

The authors propose an effective and simple way to increase the resolution of the visual part of VLM to increase the accuracy of the VLM, splitting the images into patches, separately applying ViT-CLIP to them, collecting (mosaic) the feature maps into a single high-resolution feature map as whole-image feature, and using residual connections to embed this feature map into the LLM.
StackFormer outperforms its baseline model LlaVA on both VQAv2, GQA, POPE  as well as on Text-Oriented and Zero-shot Video QA benchmarks.
StackFormer achieves best performance when the backbone is fine-tuned, while when fine-tuning a backbone without a StackFormer, the improvement is limited.

**Weaknesses:**

In Fig. 1 and 2 are missing details about the implementation of StackFormer:
- how exactly you split the high-resolution image into patches
- and how exactly you split the high-resolution visual tokens into different token sets with spatial dilation

While you write that StackFormer achieves the best trade-off between performance and effectiveness without introducing extra visual tokens, specific indicators of overhead costs when using Stackformer are not provided. Flops, parameters, latency, memory consumption, accuracy are not provided in a single table to compare Stackformer with other VLMs.

**Questions:**

Can you show in more detail in the Figures 1 and 2:
- how exactly you split the high-resolution image into patches?
- and how exactly you split the high-resolution visual tokens into different token sets with spatial dilation?

Can you provide numerical indicators (flops, params, latency, memory consumption) that Stackformer achieves the best trade-off between performance and effectiveness compared to other VLMs?

**Limitations:**

Limitation and Future Work:
The paper presents limited options for processing high-resolution images by naively splitting the image into many patches and applying VIT-CLIP to them separately. While there are many approaches to processing high-resolution images using transformers: ViTDet [1], SwinV2 [2], Patch-Fusion [3], ... or simply naively resize the ViT-CLIP model to the required image resolution, using 2D interpolation of the pre-trained position embeddings [4].


1. Exploring Plain Vision Transformer Backbones for Object Detection, Yanghao Li, Hanzi Mao, Ross Girshick, Kaiming He, 2022
2. Swin Transformer V2: Scaling Up Capacity and Resolution, Ze Liu, Han Hu, Yutong Lin, Zhuliang Yao, Zhenda Xie, Yixuan Wei, Jia Ning, Yue Cao, Zheng Zhang, Li Dong, Furu Wei, Baining Guo, 2021
3. PatchFusion: An End-to-End Tile-Based Framework for High-Resolution Monocular Metric Depth Estimation, Zhenyu Li, Shariq Farooq Bhat, Peter Wonka, 2023
4. An Image is Worth 16x16 Words: Transformers for Image Recognition at Scale, Alexey Dosovitskiy, Lucas Beyer, Alexander Kolesnikov, Dirk Weissenborn, Xiaohua Zhai, Thomas Unterthiner, Mostafa Dehghani, Matthias Minderer, Georg Heigold, Sylvain Gelly, Jakob Uszkoreit, Neil Houlsby, 2020

---

> ### Author Rebuttal · Authors · 2024-08-07
>
> We provide a detailed architecture figure in our **rebuttal pdf**. We recommend referring to Figure. 1 and Figure. 2 in our **rebuttal pdf** for a better understanding of the high-resolution token processing.
>
> ## q1-2: How to split high-resolution images into patch crops; and how to split high-resolution visual tokens into different sets.
>
> Please refer to **global author rebuttal**, and Fig.1 and Fig.2 in our **rebuttal pdf**.
>
> ## q3: Comparison of flops, params, latency
>
> Thanks for the great suggestion! We compare our StackFormer with LLaVA-1.5 and another representative work VILA [1]. As shown in the table, our model improves the baseline by introducing a ~3% increase in FLOPs without significantly increasing the iteration time during training. Additionally, our StackFormer does not require extra data or the intermediate stage training used in VILA while achieving comparable performance, which is of great training efficiency. We will include the analysis and comparison in our revision.
>
> **Comparison on 7B models**
>
> |Method | Training cost | Training Data (PT+SFT) | iter time |TFLOPs | Params |SEED | TextVQA | DocVQA | ChartQA | InfoVQA |
> | ----| ---- | ---- |---- |---- |---- | ---- |---- |---- |---- |---- |
> LLaVA-1.5 | ~0.1k GPU hours | 558K+665K | 19.0 | 26.7 | 7.06B | 58.6 | 58.2* | 28.1 | 18.2 | 25.8
> VILA-1.5  $\star$ |~5k GPU hours | 50M+1M| 17.2 | 26.7 |  7.06B | 61.1 | 64.4* | 44.5* | 25.7 | 32.5 |
> LLaVA-1.5+StackFormer | ~0.1k GPU hours| 558K+665K  | 17.9 | 27.5 | 7.06B |  60.6 | 62.4* | 39.1 | 19.2 | 29.8 |
> LLaVA-1.5+StackFormer $\star$ | ~0.1k GPU hours | 558K+665K | 18.3| 27.5 |7.06B | 63.3 | 64.5* | 39.3 | 21.0 | 30.1 |
>
> *For iter time, we average the time cost of 100 training iterations under the same 8xA100 machine*.
> *$\star$ indicates that the vision encoder is fine-tuned during the SFT stage*.
> *\* indicates that the images of the benchmark training set are observed during SFT stage*.
>
> [1] VILA: On Pre-training for Visual Language Models
>
> ## q4: Other approaches to processing high-resolution images
>
> Thanks for sharing different works in the related domain. In this work, our primary goal is to efficiently and effectively process image tokens extracted from high-resolution images for **large multimodal models (LMMs)**, rather than the techniques on how to extract high-resolution features. Therefore, we employ a simple and widely used approach, *i.e.*, adaptive multi-crop, for high-resolution feature extraction to ensure a fair comparison with LlaVA and other works.
>
> To further comprehend our findings, we follow the suggestion and conduct experiments using other approaches to process high-resolution images, as follows.
>
> 1. [*Our default setting*] Multi-crop sub-images + StackFormer-LLM (freeze ViT).
>
> 2. Multi-crop sub-images + StackFormer-LLM (unfreeze ViT): We unfreeze the parameters of ViT during the SFT stage, building on the approach in #1.
>
> 3. Multi-crop sub-images + StackFormer-ViT (unfreeze ViT): We use the first 20 layers of ViT to extract multi-crop high-resolution features and the last 4 layers to stack these features. The stacked features are then fed into a projection module as visual token inputs for LMMs.
>
> 4. whole image, 2D interpolation on ViT + StackFormer-LLM (unfreeze ViT): We directly interpolate the positional embedding of ViT to process high-resolution images. StackFormer is then used to stack the extracted features. ViT needs to be unfrozen because the input resolution differs from the pre-training vision encoder.
>
> 5. whole image, ViT-det + StackFormer-LLM (unfreeze ViT): We apply techniques from ViTDet, incorporating global and local attention in transformer blocks and discarding the CLS token. StackFormer is used for token stacking in the LLM.
>
> We conduct ablation experiments using Phi-3 (3B) as our language model. As shown in the table below, our StackFormer on LLM/ViT (#1, #2, #3) achieves significant performance gains compared to the baseline model. However, other methods, such as 2D positional interpolation and ViT-Det, result in performance drops due to inconsistencies between image pre-training and the LMM SFT stage. These techniques commonly used on detection often require longer training schedules (*e.g.*, 100 epochs on COCO for ViTDet), which may be unsuitable for the one-epoch SFT pipeline used in LMMs.
>
> #### Ablations on processing high-resolution images with phi-3 (3b LLM)
> | # |Method | AVG | SEED | TextVQA | DocVQA | ChartQA | InfoVQA
> |---- | ---- |---- |---- |---- | ---- |---- |---- |
> |0 |Baseline | 38.1 | 62.6 | 55.7 | 28.1 | 15.8 | 28.3
> |1 |Multi-crop, Stackformer-LLM | 40.9 | 62.9 | 58.4 | 37.8 | 16.6 | 28.9 |
> |2 |Multi-crop, Stackformer-LLM (unfreeze ViT) | 42.5 | 63.9 | 60.3 | 39.0 | 19.1 | 30.1
> |3 |Multi-crop, Stackformer-ViT | 42.0 | 64.0 | 60.1 | 38.4 | 17.1 | 30.6
> |4 |Whole image, 2d pos-interpolation, Stackformer-LLM| 36.1 |62.7 | 53.4 | 25.1 | 14.8 | 24.5
> |5 |Whole image, ViT-Det style, Stackformer-LLM  | 34.1 | 60.3 | 48.6 | 22.6 | 14.2 | 25.0
>
> Furthermore, we conduct additional experiments to utilize StackFormer on LLM/ViT (#1, #2, #3) with 7/13B LLM models, which further demonstrate the effectiveness of our StackFormer. Please refer to part 3 (Additional Results on vicuna-1.5) in **global author rebuttal**.
>
> We believe that other techniques for extracting high-resolution features with vision transformers, such as patch-fusion [1] and swin-attention [2], may have the potential to obtain higher performance than the commonly-used naive multi-crop approach. We leave these explorations in the future work.
>
> [1] PatchFusion: An End-to-End Tile-Based Framework for High-Resolution Monocular Metric Depth Estimation.
> [2] Swin Transformer V2: Scaling Up Capacity and Resolution.

---

> > ### Comment · Reviewer_REjR · 2024-08-12
> >
> > Thanks to the authors for the answers, detailed schemes and additional results. This makes the approach clearer. Based on this, I increase the rating of the paper to "6: Weak Accept"

---

### Official Review · Reviewer_SQxC · 2024-07-13

**Soundness:** 4
**Presentation:** 2
**Contribution:** 4
**Rating:** 7
**Confidence:** 5

**Summary:**

This paper proposes a new visual token organization method. Specifically, it proposes to stack visual tokens instead of the commonly used stringing. Experiments show that the proposed StackFormer can improve performance on TextVQA, DocVQA, and InfoVQA.

**Strengths:**

- The proposed method is novel, different from the commomly-used stringing method.
- Experiments show that the proposed method StackFormer can significantly improve performance on some datasets, especially the traditional VQA datasets.

**Weaknesses:**

- There are many typos in the paper. The authors need to improve their writing and polish the paper. Like Line 18 "StackFormeruses" --> "StackFormer uses"; Line 134: Multi-modal Language Models (LLMs). Sometimes it uses LMMs and sometimes MLLMs. Both are ok, but please use only one in the same paper.

- There is no significant improvement for LLaVA-Next on MLLM benchmarks.

**Questions:**

- In table 1, what do † and * mean?

- Could you explain why StackFormer cannot improve LLaVA-NeXt on MLLM benchmarks?

**Limitations:**

the authors have discussed the limitations in the paper. StackFormer cannot significantly improve larger model LLaVA-NeXt on MLLM benchmarks. It may be another limitation.

---

> ### Author Rebuttal · Authors · 2024-08-07
>
> ## q1: Typo
> Thank you for pointing it out, we have polished the representation and will update it in the next version.
>
> ## q2: Improvement for LLaVA-Next on MLMM benchmarks
> Thank you for your valuable comments! We suspect that two main factors contribute to the results: (1) the image resolution, and (2) the quality of the SFT dataset (please refer to L227-L228 in our main paper).
>
>
> LLaVA-Next scales up the effective resolution from $336 \times 336$ to {$672\times 672$, $336\times 1344$, $1344 \times 336$} by using dynamic multi-image crops and stringing visual tokens [1]. This approach allows the model to handle most resolutions in MLLM benchmarks as mentioned. Quantitatively, we analyzed the average width and height of images from each benchmark. As shown in the table below, the average resolution of multi-modal benchmarks is significantly smaller than that of text-oriented benchmarks. Consequently, the multi-image crops already suffice to cover most of those benchmarks in LLaVA-Next. Further upsampling the image to a higher resolution using bilinear interpolation does not introduce additional information. As a result, the proposed layer stacking of visual tokens extracted from upsampled images does not necessarily show gains on them as significant as the originally high-resolution text-oriented benchmarks.
>
> |Benchmark | average resolution (height $\times$ width) |
> | ----| ---- |
> TextVQA | 819.4 $\times$ 952.3
> DocVQA | 2098.6  $\times$ 1782.8
> InfoVQA | 3002.4  $\times$ 1161.5
> SEED | 899.2  $\times$ 1090.0
> POPE | 478.8  $\times$ 584.7
> MMMU | 488.2  $\times$ 723.0
> MMVet | 797.4  $\times$ 1059.5
>
> As discussed in our main submission, the SFT-768K data used in LLaVA-Next includes a portion of private data collected from the LLaVA demo. This part of the data is used to generate responses with GPT-4V to obtain high-quality instruction-following data. In contrast, our SFT-748K dataset lacks this component, leading to limitations on the GPT4-evaluated benchmark, *i.e.* MM-Vet.
>
> Furthermore, we observe that MLLM benchmarks emphasize evaluating reasoning, hallucination, *etc.*, while text-oriented benchmarks prioritize fine-grained perception and understanding (please refer to Fig. 3 in **our rebuttal PDF**). Additionally, previous works [2] indicate that MLLM benchmarks, such as MMMU, benefit more from LLM capabilities rather than visual representations. We suggest these differences may contribute to that StackFormer for LLaVA-Next does not obtain significant gains compared to text-oriented benchmarks.
>
>
> ## q3: In table 1, what do † and * mean?
> In Table 1, † indicates that the model is continuously fine-tuned based on the LLaVA-Next checkpoint. Since the training codes and the 765K instruction tuning data used in LLaVA-Next are not publicly available, we mix a 748K SFT dataset based on information from the LLaVA-Next blog. However, we do not have access to the private data used in LLaVA-Next or the exact combination ratios among the datasets. Consequently, we fine-tuned our model using our 748K dataset and LLaVA-Next to ensure a fair comparison.
>
> Additionally, * denotes that the training images from the downstream benchmark are observed during the SFT stage. Besides, we mark MMMU* to indicate that we are reporting the validation results for MMMU. We will clarify these details in the next version of our paper.
>
> [1] LLaVA-NeXT: Improved reasoning, OCR, and world knowledge
> [2] LLaVA-NeXT: What Else Influences Visual Instruction Tuning Beyond Data?

---

> > ### Comment · Reviewer_SQxC · 2024-08-12
> >
> > Thanks for the rebuttal. I would like to keep my score of 7: Accept.

---

### Official Review · Reviewer_QJM9 · 2024-07-19

**Soundness:** 3
**Presentation:** 4
**Contribution:** 3
**Rating:** 7
**Confidence:** 4

**Summary:**

This paper proposes a method to add more visual information to a MM-LLM without increasing the number of tokens processed by the model. The idea is simple, just add visual tokens to the existing hidden representation between each layer of the transformer. The approach is evaluated on many tasks and shows good results.

**Strengths:**

The paper is well written, the approach would be reproducible from the given descriptions. The idea is novel and simple and effective. The experiments are thorough.

**Weaknesses:**

Overall the paper is well done. The experiments are thorough, the idea is well explained, and the method is reproducible.

**Questions:**

None

---

> ### Author Rebuttal · Authors · 2024-08-07
>
> Thank you for your comments! We are pleased that you find our work novel and simple! If you have any additional questions, please feel free to add detailed comments; we are happy to answer them and sincerely hope to address your concerns if any.

---

### Author Rebuttal · Authors · 2024-08-07

First of all, we sincerely appreciate all your valuable comments and suggestions.

In this work, we proposed a new model called Stackformer to handle the long sequence of visual tokens in large multimodal models (LMMs). Unlike all previous works that string visual tokens into a long sequence, we instead stack them layer by layer and feed them into the large language models using a simple residual connection. As demonstrated by our extensive empirical studies, the proposed method significantly improves the LMMs' ability to handle high-resolution images while keeping the context length unchanged for LLMs.

We are encouraged that all reviewers gave positive ratings to our work and recognize the merits of our work including:

* **Novelty**: "The idea is novel" (R-QJM9), "The proposed method is novel, different from the commonly-used stringing method" (R-SQxC).
* **Simplicity**: "simple and effective" (QJM9), "The authors propose an effective and simple way" (REjR).
* **Effectiveness**: "simple and effective" (QJM9), "can significantly improve performance on some datasets" (SQxC), "The authors propose an effective and simple way" (REjR).

We carefully read the comments by all reviewers and attempted to provide comprehensive responses accordingly. Please find the rebuttal below each official review. We hope the responses could answer the questions raised by reviewers and address any concerns about our work.

Thanks again to all reviewers for the time and effort!


***


## Author Global Responses

We provide a detailed architecture figure in our **rebuttal pdf**. We recommend referring to Figure. 1 and Figure. 2 in our **rebuttal pdf** for a better understanding of the high-resolution token processing.

### 1. How to split high-resolution images into patch crops
We split high-resolution images into sub-image crops using dynamic high-resolution techniques [1,2,3]. Given the grid pinpoints template $A$={($a^{h}_i, a^{w}_i$)} and the input resolution of vision encoder $\mathrm{r}$, the resolution candidates are calculated as $R$={($r \cdot a^{h}_i, r \cdot a^{w}_i$)}. For each image $I$, we first select the best-fitting resolution from $R$ to resize $I$. The resized image is then split into fixed-size sub-images of $r \times r$ accordingly.

For example, we use CLIP-ViT-336 as the image encoder and define the grid pinpoints templates $A$ = {(1,2), (1,3), (1,4), (2,1), (2,2), (3,1), (4,1)} to ensure at most 4 sets of high-resolution tokens for stacking by default. This setup allows us to obtain the resolution candidates $R$ = {(336, 336), (336, 672), (336, 1008), (336, 1344), (672, 336), (672, 672), (1008, 336), (1344, 336)}. For each input image, we first select the best-fitting resolution. The image is then resized and split into $336 \times 336$ sub-images accordingly. Thus, the high-resolution sub-images can be directly encoded with the image encoder.

Please refer to Fig. 1 in our **rebuttal pdf** for a clearer understanding.

[1] SPHINX: The Joint Mixing of Weights, Tasks, and Visual Embeddings for Multi-modal Large Language Models.
[2] SPHINX-X: Scaling Data and Parameters for a Family of Multi-modal Large Language Models.
[3] LLaVA-NeXT: Improved reasoning, OCR, and world knowledge.

### 2.  How to split high-resolution visual tokens into different sets
Given the high-resolution visual tokens $\mathbf{X}^{hires} \in \mathbb{R}^{(a^h \cdot h) \times (a^w \cdot w) \times C}$, we apply 2D sampling to divide the tokens into different sets. Here, $h$ and $w$ represent the image feature shape of the vision encoder, while $a^h$ and $a^w$ denote the aspect ratio of the resized image. The 2D sampled token sets for StackFormer are calculated as $\mathbf{X}^{stack} = \{\mathbf{X}^{hires}[i::a^{h}, j::a^{w}]\}$, where $i$ = {$0, 1, \ldots, a^{h} - 1$} and $j$ = {$0, 1, \ldots, a^{w} - 1 $}.

Please refer to Fig.2 in our **rebuttal pdf** for a better understanding.


### 3. Additional Results on vicuna-1.5 (7b/13b LLM)
Inspired by Reviewer REjR, we further conduct experiments on 7b and 13b LMMs. For StackFormer-ViT, we use the first 20 layers of ViT to extract multi-crop high-resolution features and the last 4 layers to stack these features. The stacked features are then fed into a projection module as visual token inputs for LMMs, without creasing the visual context length.

The results demonstrate that our StackFormer can be effectively utilized for both LLM and ViT.

*$\star$ indicates that the vision encoder is fine-tuned during SFT stage.*
| # | Method |  VQAv2 | GQA | TextVQA | DocVQA | InfoVQA | SEED | POPE | MMMU | MM-Vet |
|---- |---- | ---- |---- |---- |---- |---- |---- |---- |---- |---- |
|0 | Llava-1.5-7b |  78.5 | 62.0 | 58.2 | 28.1 | 25.8 | 58.6 | 85.9 | 35.3 | 30.5
|1 | Stackformer-LLM-7b   | 79.5 | 63.1 | 62.4 | 39.1 | 29.8 | 60.6 | 86.7 | 35.7 | 29.9
|2 | Stackformer-LLM-7b $\star$   | 81.1 | 63.9 | 64.5 | 39.3 | 30.1  | 63.3 | 86.7 | 37.1 | 29.8
|3 | Stackformer-ViT-7b $\star$  | 80.4 | 64.1 | 63.5 | 41.0 | 30.0 | 62.3 | 87.6 | 34.9 | 33.0

| #| Method | VQAv2 | GQA | TextVQA | DocVQA | InfoVQA | SEED | POPE | MMMU | MM-Vet |
---- |---- | ---- |---- |---- |---- |---- |---- |---- |---- |---- |
0 | Llava-1.5-13b  | 80.0 | 63.3 | 61.3 | 30.3 | 28.4 | 61.6 | 85.9 | 35.3 | 30.5
1 |Stackformer-LLM-13b   | 80.9 | 64.2 | 64.6 | 41.5 | 33.0 | 63.5 | 87.7 | 35.2 | 35.9
2 |Stackformer-LLM-13b $\star$  | 82.1 | 65.1 | 65.2 | 43.1 | 34.0 | 64.4 | 86.6 | 34.7 | 36.2
3 |Stackformer-ViT-13b $\star$  | 81.1 | 64.2 | 63.9 | 41.7 | 33.1 | 63.0 | 86.6 | 34.7 | 31.1

---

### Decision · Program_Chairs · 2024-09-25

**Decision:**

Accept (poster)

**Comment:**

This paper has received consistent feedback from three reviewers, all of whom are in favor of acceptance. The paper proposes a simple yet effective layer stacking strategy to significantly increase the number of tokens in large language models (LLMs). The experiments are comprehensive, and the results are solid. Therefore,  the AC has decided to accept this paper.